# Alkaloids with Anti-Onchocercal Activity from *Voacanga africana* Stapf (Apocynaceae): Identification and Molecular Modeling

**DOI:** 10.3390/molecules26010070

**Published:** 2020-12-25

**Authors:** Smith B. Babiaka, Conrad V. Simoben, Kennedy O. Abuga, James A. Mbah, Rajshekhar Karpoormath, Dennis Ongarora, Hannington Mugo, Elvis Monya, Fidelis Cho-Ngwa, Wolfgang Sippl, Edric Joel Loveridge, Fidele Ntie-Kang

**Affiliations:** 1Department of Chemistry, Faculty of Science, University of Buea, P.O. Box 63, Buea CM-00237, Cameroon; ajeck.james@ubuea.cm; 2AgroEco Health Platform, International Institute of Tropical Agriculture, Cotonou, Abomey-Calavi BEN-00229, Benin; 3Institute for Pharmacy, Martin-Luther-Universität Halle-Wittenberg, Kurt-Mothes-Str. 3, 06120 Halle, Germany; veranso.conrad@gmail.com (C.V.S.); wolfgang.sippl@pharmazie.uni-halle.de (W.S.); 4Department of Pharmaceutical Chemistry, School of Pharmacy, University of Nairobi, Nairobi P.O. Box 19676–00202, Kenya; koabuga@gmail.com (K.O.A.); dbagwasi@gmail.com (D.O.); mugohannington@yahoo.com (H.M.); 5Department of Pharmaceutical Chemistry, School of Chemistry, University of KwaZulu-Natal, Durban 4001, South Africa; karpoormath@ukzn.ac.za; 6ANDI Centre of Excellence for Onchocerciasis Drug Research, Biotechnology Unit, Faculty of Science, University of Buea, P.O. Box 63, Buea CM-00237, Cameroon; munyane001@gmail.com (E.M.); fidelis.cho@ubuea.cm (F.C.-N.); 7Department of Chemistry, Swansea University, Singleton Park, Swansea SA2 8PP, UK; 8Institute of Botany, Technical University of Dresden, 01217 Dresden, Germany

**Keywords:** alkaloids, anti-onchocercal, bisindoles, docking, homology modeling, monoindoles

## Abstract

A new iboga-vobasine-type isomeric bisindole alkaloid named voacamine A (**1**), along with eight known compounds—voacangine (**2**), voacristine (**3**), coronaridine (**4**), tabernanthine (**5**), iboxygaine (**6**), voacamine (**7**), voacorine (**8**) and conoduramine (**9**)—were isolated from the stem bark of *Voacangaafricana*. The structures of the compounds were determined by comprehensive spectroscopic analyses. Compounds **1**, **2**, **3**, **4**, **6**, **7** and **8** were found to inhibit the motility of both the microfilariae (Mf) and adult male worms of *Onchocerca ochengi*, in a dose-dependent manner, but were only moderately active on the adult female worms upon biochemical assessment at 30 μM drug concentrations. The IC_50_ values of the isolates are 2.49–5.49 µM for microfilariae and 3.45–17.87 µM for adult males. Homology modeling was used to generate a 3D model of the *O. ochengi* thioredoxin reductase target and docking simulation, followed by molecular dynamics and binding free energy calculations attempted to offer an explanation of the anti-onchocercal structure–activity relationship (SAR) of the isolated compounds. These alkaloids are new potential leads for the development of antifilarial drugs. The results of this study validate the traditional use of *V. africana* in the treatment of human onchocerciasis.

## 1. Introduction

*Voacanga africana* Stapf (Apocynaceae) has been applied extensively in traditional medicine, particularly in Africa. For example, the root bark of the plant is used to treat diarrhea in Kinshasa, the Democratic Republic of Congo [1], while the stem bark has been used in combination with the sister species *V. thouarsii* in the treatment of heart disease, leprosy, diarrhea, generalized oedema, madness and convulsions in children in Ivory Coast, Cameroon, Ghana and Congo [2,3]. Additionally, the fruits, bark and leaf extracts have been used in Cameroonian ethnomedicine to treat orchitis, ectopic testes and gonorrhoea, respectively [4,5,6,7,8]. Of late, Apocynaceous plants have been under scientific investigation. Monoterpenoids and bisindole alkaloids are the major classes of secondary metabolites isolated from this plant family [9,10,11,12]. Recently, our research group carried out investigations that led to the identification of the alkaloids voacangine (**2**) and voacamine (**7**), from the stem bark of *V. africana*, with both compounds showing activity against *Onchocercaochengi* [13]. The cattle derived *O*. *ochengi* is the best model and closest known relative of *O*. *volvulus* based on phylogenetic relationship [14]. This was the first report of anti-onchocercalactivity of alkaloids from this plant species. To the best of our knowledge, the majority of the isolated compounds from this plant family have not been screened against onchocerciasis, although the disease still stands as a major public health problem in the developing world [15]. Thus, as part of our research program aimed at investigating new bioactive compounds in anti-onchocercal drug discovery, the present study was undertaken to identify novel inhibitors from this plant using chemical and molecular modeling methods in an attempt to explain their structure–activity relationships.

## 2. Results and Discussion

### 2.1. Isolation and Identification of Alkaloids

The crude methanol extract of *V*. *africana*was subjected to silica gel normal phase open column chromatography and elution with a gradient of ethyl acetate in hexane. Repeated column chromatography through Sephadex LH-20, preparative thin-layer chromatography (TLC) yielded a new bisindole alkaloid derivative named voacamine A (**1**) along with eight known compounds—voacangine (**2**), voacristine (**3**), coronaridine (**4**), tabernanthine (**5**), iboxygaine (**6**), voacamine (**7**), voacorine (**8**) and conoduramine (**9**) [16,17,18,19,20,21,22,23,24,25] (Figure 1). The structures of the compounds were established based on NMR (Appendix A) analyses as well as by comparison with published data. For **1**, infrared and mass spectra (Appendix A), and optical rotation, were also acquired.

Compound **1**, named voacamine A, was obtained as a cream-colored powder, with a melting point of 222–223 °C and α_D_−37° (c = 0.13, CHCl_3_) through crystallization from methanol. This showed a positive reaction with Dragendorff’s reagent. The compound was identified by comparing it with an authentic sample of voacamine based on its high-performance liquid chromatography (HPLC) retention times. The UV absorption bands at 228, 287, 295 and 308 nm indicated **1** to be an indole alkaloid. The major IR absorptions occurred at 3383 cm^−1^ (indole NH), 2930 and 2856 cm^−1^(C-H stretches), 1722 cm^−1^ (ester), 1463 cm^−1^ (indole), 1208 and 739 cm^−1^, and were in line with those expected for bisindole alkaloids. High resolution mass spectrometry revealed a major ion at *m*/*z* 705.4022 ([M + H]^+^, calc. 705.4016, C_43_H_53_N_4_O_5_).

The ^1^H- and ^13^C-NMR data of **1** were characteristic of a bisindole alkaloid (Table 1), with almost all chemical shifts very similar to those published for voacamine [18]. The ^13^C-NMR data showed a total of 43 resonances, including those at *δ* 171.5 (ester carbonyl), 49.9 (ester methyl), 42.4 (N-Me), 118.8, 137.8 (olefinic C-19, C-20) on the vobasinyl subunit, and 11.6, 26.7 (C-18′, C-19′, iboga). The ^1^H-NMR data of **1** showed the presence of two indole NH groups (*δ* 7.49, 7.74, 1H each), an unsubstituted indole moiety (*δ* 7.07–7.57, 4H, vobasinyl); another, disubstituted, indole ring (*δ* 6.76 and 6.95, 1H each, iboga); an aromatic methoxy group (*δ* 4.03, 3H, iboga); two methyl esters (*δ*3.67, 2.48, each 3H); a *N*-methyl group (*δ* 2.63, 3H); and an ethylidene side chain (*δ* 1.69, 3H and 5.36, 1H) on the vobasinyl scaffold. ^15^N-HSQC and ^15^N-HMBC (Appendix A) analyses showed no evidence of either subunit being an N-oxide, as the ^15^N chemical shifts were inconsistent with this functional group, and neither showed any evidence of a hydroxyl group on C-19’, as is seen in **8** and **9**.

Notably, all ^1^H and ^13^C chemical shifts for the iboga subunit were essentially identical to those observed for **7** [18], allowing us to assign the indole substitution pattern of that moiety to be the same as observed in **7** and **8**, and demonstrating that the vobasinyl moiety is linked to C-11′ of the iboga moiety. The upfield chemical shift of C-9′ (*δ* 99.2) is a characteristic of adjacent (C-10′) oxygenation and confirms the presence of an aromatic methoxy group at C-10′ of the iboga subunit. Furthermore, in the NOESY and HMBC (Figure 2) spectra, a H-9′/Ar-OMe NOE and the three-bond correlation from MeO-10′ to C-10′ were observed.

The ^1^H and ^13^C chemical shifts for the vobasinyl subunit were also extremely similar to those of voacamine, except for a significant difference in the chemical shifts of C-2, C-20, H-16, H-19 and one each of the H-6 and H-21 methylene pairs. The broad resonance at *δ* 5.17 was assigned to H-3 on the vobasinyl subunit, which allowed us to confirm linkage of the iboga portion of the bisindole alkaloid to C-3 of the vobasine scaffold, as no alternative linkage point could be found. NOEs to this H-3 resonance were similar in **1** to in **7** (except where noted below), indicating that the same stereochemistry is observed at C-3.

In addition to the chemical shift differences, the NOESY (Appendix A) data show key structural differences between **1** and **7**. In **7**, H-15/H-18 and H-19/H-21 NOEs are observed [18], whereas for **1** these are not seen, and instead H-18/H-21 and H-15/H-19 NOEs are observed, showing that the alkene geometries are different. Similarly, **7** gives NOEs from H-16 to the adjacent H-5 and H-15 only, whereas **1**, in addition to these, gives a clear H-3/H-16 NOE, which indicates that the methyl ester must have the opposite orientation, and, therefore, C-16 the opposite stereochemistry to that observed in **7 [18]**. The H-3/H-16 NOE also provides further evidence that H-3 and C-16 are on the same face of the vobasinyl subunit. The stereochemistry at C-20′ in the iboga subunit is, however, identical to that in **7**, as evidenced by H-3′/H-19′ and H-17′/H-20′ NOEs. The structural differences between **1** and **7** also explain the chemical shift differences: C-2 and one H-6 are under the methyl ester in **7**, whereas C-20 and one H-21 are under it in **1**, and the different geometry of the ethylidene group places H-19 in a different environment. The different stereochemistry at C-16 is known (in *epi*-voacamine), but bisindole alkaloids with a different alkene geometry, to the best of our knowledge, are not. The structure of **1** was, therefore, determined as voacamine A.

### 2.2. Anti-Onchocercal Activity

The isolated compounds were screened for their anti-onchocercal activities against *O*. *ochengi* using methods described in the literature [26,27]*. O*. *ochengi*is the best model and closest relative of *O. volvulus* that is cheap and easily available for performing research in chemotherapy and immunology of onchocerciasis. Primary screening was performed to eliminate inactive compounds. Seven of the isolates showed activity against *O. ochengi* worms in primary screens. Compounds **1**–**4** and **6**–**8** were found to inhibit the motility of both the microfilariae (Mf) and adult male worms of *O*. *ochengi*, in a dose-dependent manner, but were only moderately active on the adult female worms upon biochemical assessment at 30 μM drug concentrations (Table 2). Their IC_50_ values were 2.49–5.49 µM for microfilariae, and 3.45–17.87 µM for adult males (Table 3).

### 2.3. Molecular Modeling of Secondary Metabolites

#### 2.3.1. Computation of Pharmacokinetic-Related Properties

Several physicochemical properties of the identified compounds were computed, particularly those related to drug metabolism and pharmacokinetics, using the QikProp software (Schrodinger 2018). The obtained values were compared with values reported for the same parameters for 95% of known drugs. Selected computed values are shown in Table 4. The number of properties for which the computed values for compounds **1** to **9** fall outside the recommended range for 95% of known drugs are marked as #stars. For 5 of the identified compounds (**2** to **6**), the values of #stars were reported as zero. This implied that all the computed properties related to drug metabolism and pharmacokinetics (DMPK) for these compounds (only the monoindoles) fell within the required range for 95% of known drugs.

Regarding the compounds with properties falling out of the required range, the bisindoles (compounds **1**, and **7**–**9**), the properties that fell outside the required range (above 1000.0 Å^2^) are marked with an asterisk in Table 4. For example, these compounds show particularly high solvent accessible surface areas, which would not be ideal for crossing the cell membrane if administered as drugs [28]. A highly polar surface will not easily penetrate the cell membranes and would require special transporters, thereby leading to a low concentration of the compound in the cells. The same trend holds for the hydrophobic component of the SASA of the same compounds (above 750 Å^2^). In this case, the higher hydrophobic surface plays a negative role in the interaction of these molecules with water (a polar solvent), hence lower solubility. The logS value is expected to range from −6.5 to 5 for most drugs. In the case of these compounds, the computed aqueous solubilities are weak (all below −6.5), implying that the only hope of them being developed into drugs would be by the introduction of polar functional groups, which could render them less hydrophobic and more polar, or to fragment them into monoindoles. The higher hydrophobicities of the bisindoles is also demonstrated in their higher logP values (all above the required maximum of 5 for orally absorbed drugs according to Lipinski’s rules [29]). The same trend is observed for the molar volumes (all bisindoles with volumes higher than the maximum of 2000 Å^3^ for the majority of known drugs). Compounds known to block human-ether-a-go-go K^+^ channels have a high tendency to provoke cardiac arrhythmia and could be fatal [30,31]. Thus, computed logHERG values are often used as indicators of toxicities of compounds. In the case where this value is less than −5, the compound is not encouraged to be further developed. This is the case of the bisindoles, all computed logHERG values were less than −5. This, coupled with the low MDCK permeabilities (an indicator of poor drug permeability, with most of the bisindoles showing less than or almost equal a value of 25, signifying weak permeability [32]), and higher number of expected metabolic reactions (10 reactions each, signifying their instability in the body) all indicate that the bisindoles are not suitable for further drug development. In addition, they have higher binding affinities to plasma proteins. In this case, the predicted binding affinities to human serum albumin (HSA) were higher than the expected maximum. In general, binding to plasma proteins limits the amount of the drug to be distributed in general circulation [33]. Furthermore, only these bisindoles violated the general rules of thumb (“Rule of Five” and “Rule of Three”), which are general guidelines for selecting orally available drugs.

#### 2.3.2. Homology Modeling

##### Model Development

To date, there is no available protein structure for the thioredoxin reductase of *O. ochengi* in the Protein Data Bank [34]. So, a multiple sequence alignment to check the identified conserved moieties for the homologous protein sequences was performed. The percentage sequence identity and similarity values to our target, compared with templates, are shown in (Appendix A). The ligand present in the template structure was transferred to the target during the homology model development process via a slight modification of the modeler script. The structural template 4JNQ_A with a dihydroflavine-adenine dinucleotide (to define and construct the binding pocket) was selected and used for the generation of homology models. Estimation of model quality was an important aspect in selecting the best model among the 100 models that were built. The final model (Figure 3) was selected based on the DOPE scoring function of the modeler program, used in the generation of the models. The Ramachandran plot (*φ*/*ψ*) distribution of the backbone conformation angles for each of the residues of the refined structure revealed that ~98.0% and ~2.0% were expected in the favored and allowed regions (Appendix A). The choice of model was considered as satisfactory and reliable to advance our study.

##### Evaluation of the Stability of the Generated Homology Model

To fully explore the generated model and use it for further in silico procedures, the model was prepared and subjected to two energy minimization steps. A 20 ns molecular dynamic simulation run using the Amber software was used to evaluate the stability of the resulting homology model [35]. Visual inspection of the trajectory of the homology model confirmed it reached an equilibrium at ~4 ns into the molecular dynamic simulation run with an rmsd value of about 2.5. Figure 4 shows that the rmsd plot of the backbone heavy atoms of the conformations sampled during the simulation time was stable.

#### 2.3.3. Docking

To explain the observed experimental reported activities of the isolated indole alkaloid, an attempt was made to understand their binding mode and interactions with the thioredoxin reductase model generated for *O. ochengi*. Docking scores and estimated minimization energies of the protein–ligand complexes for the compounds docked into the target site of *O. ochengi* thioredoxin reductase, using rigid ligand docking within the active site of the generated model, are tabulated (Table 5). The docking poses of the ligands within the protein binding site in the putative binding mode are shown in Figure 5.

#### 2.3.4. Binding Free Energy Calculations

To further explore the SAR, the free energy of binding of each ligand to the receptor in the putative binding mode was calculated, along with its electrostatic (E_ele_), van der Waals (E_vdw_) and solvation (E_sol_) components. These are included in Table 6.

Compound **8** was not included in the calculation, since both the vobasinyl and the iboga moieties of this compound are represented in the other structures, e.g., the vobasinyl moiety of compound **8** is the same as that of compounds **7** and **9**, i.e., **7a**, while the iboga moiety of compound **8** is the same as that of compound **9**. The computed ΔG_bind_ energy values showed the lowest binding affinity (tightest binding) for compound **7** towards the target (Table 6), with no clear-cut differences in binding affinities of the monoindoles towards the targets, when compared with the bisindoles. Amongst the monoindoles (compounds **2** to **6**), compounds **2** and **3** showed much tighter binding compared to compounds **3** to **6**. Meanwhile, among the bisindoles, the lowest affinities were seen in compounds **1** and **7**. In general, the greatest contribution towards the computed binding free energies, which were approximately 2- to 5-fold that ofthe van der Waals interactions in the monoindoles and 2- and 6-fold in the bisindoles, was electrostatic. Furthermore, the computed affinities that had a greater electrostatic contribution to binding free energy led to better affinities. Since van der Waals contributions are generally more important for hydrophobic molecules while the electrostatic contribution to binding affinity is more important for polar compounds, it was clear that the introduction of more polar groups to the monoindoles would show better prospects towards binding to the thioredoxin reductase target of *O. ochengi*.

#### 2.3.5. Structure–Activity Relationships

From Table 3, it can be observed, in terms of IC_50_ values, that:Compound **1** is more active than compound **2**, being ~2 fold more active in both Mf and adult male worms. In our discussion, the activities against adult female worms were ignored, since the experimental activities were limited to a few dotted cases.The measured activities of the bisindoles (**1**, **7** and **8**, possessing a vobasinyl unit) were much better than those of the mono indoles (**2**–**4** and **6**).Compound **5** was almost inactive in all assays and is not included in Table 3 and this discussion.

The activities of all tested bisindoles lie within the same range and are twice as active as all the tested monoindoles. Among the monoindoles, it was noted that since they lack the vobasinyl unit, they all have much weaker activities against both Mf and adult males than the bisindoles. In the continuation of our discussion, we shall refer to the vobasinyl unit of compound **1** as **1a**, while the vobasinyl unit of compound **7** would be referred to as **7a** and the iboga unit of compound **9** would be referred to as **9b**.

Among the monoindoles (**2** to **6**), the least active is compound **5**. Compound **4** is most active against Mf, while **3** is most active in adult males (Table 3). This shows that the OMe and OH groups do not play any role in the activity. On the contrary, the ester group (present in both the most active compounds **2** and **3**) is absent in the least active monoindoles (**5** and **6**). This ester group interacts with the triad (Gly116, Ala117 and Ala114) in the docking pose of compound **2** (Figure 5B), making a H-bond with Gly116. The same ester group in the iboga unit of compound **7** makes the same interaction with the same triad (Appendix A). The docking pose of compound **4** (Figure 5D) also shows the ester group interacting with the Gly116, Ala117 and Ala114 triad, which is an indication that the monoindoles that lack this ester group (compounds **5** and **6**) would miss this interaction and, hence, be less active than their counterparts (**2**–**4**) in both Mf and adult male worms. The ester group is also present in all vobasinyl units of the bisindoles, but only interacts with Asn249 in the docking pose of compound **1** (Figure 5A), contrary to compounds **2** and **3**.

Comparing the docking pose of compound **1** (Figure 5A) and its vobasinyl moiety (**1a**, Appendix A), we observe that the indole NH of compound **1** rather makes a H-bond interaction with Asp284, while the ester group interacts with the Asn249. The former is absent in the interaction between 1a and the binding site, this moiety only interacting via H-bonding with Ala114. When compared with compound **7**, the isomer of compound **1**, the vobasinyl moiety (**7a**) only interacts with Asn249 and not with the triad, while when the entire compound **7** was docked, the ester group rather interacted with the triad and not with the Asn249 (Appendix A). We must, however, note that moiety **7a** is the geometric isomer of moiety **1a**, while compound **2** is equivalent to the iboga unit of compound **7** and compound **3** is the iboga unit of compound **8**. One could also explain the much higher activities of the bisindoles when compared with the mono indoles, because the mono indoles generally lack the vobasinyl units, thus missing the right interactions in the binding site. This could be because this part of the molecule does not interact with the aforementioned residues, the two compounds (**1** and **7**) almost binding in the same way (yellow structures of Figure 5A and Appendix A).

In terms of docking scores (Table 5), all bisindoles (the most active compounds, except the untested compound **9**) had higher docking scores (−4.25, −4.25 and −4.72 kcal/mol) for compounds **1**, **7** and **8**, respectively, towards the docked receptor. When compared with the monoindoles (ranging from −4.79 to −5.26 kcal/mol), the docking scores of the monoindoles are much lower. It must be noted that the stereoisomers **1** and **7** had the same docking scores (−4.25 kcal/mol) towards the receptor site, an indication that the difference in stereochemistry had no effects on the top scoring docking poses. All the indoles had higher docking scores towards the receptor site, when compared with the reference compound auranofin (−6.37 kcal/mol), but this could be explained by the fact that this compound could have a completely different binding mode or even different binding site from the reported *Voacanga* indoles. The chemical structure of auranofin (PubChem ID: 24199313) is shown in the Supplementary Data (Appendix A). A similar trend could be seen in terms of the minimization energies of the ligand–receptor complexes (Table 5), except for compound **8** (which had a much higher minimization energy (−28.41 kcal/mol), when compared with its other bisindole counterparts (**1** and **7**), with minimizations energies of −30.48 and −30.43 kcal/mol, respectively. When compared with the monoindoles (compounds **2** to **4**), the minimization energies of the complexes with bisindoles are much lower (more stable) than those with monoindoles. It was generally observed that the isolated molecules bound almost mostly via hydrophobic interactions with amino acid side-chains in the *O. ochengi* thioredoxin reductase target site. In terms of binding free energies, although no clear SAR could be derived, it was observed that electrostatic contributions towards the binding of the compounds towards the target were more important than van der Waals interactions.

## 3. Materials and Methods

### 3.1. General Experimental Procedures

Column chromatography was carried out with glass columns using Merck 60 (Merck, Darmstadt, Germany) (60–200μm) silica gel as a stationary phase. Size exclusion chromatography was performed with Sephadex LH-20 (Sigma Aldrich, Seelze, Germany). Preparative TLC was performed using silica gel H. Analytical TLC was performed on Merck F254 aluminium sheets precoated with silica gel with *n*-hexane in ethyl acetate as the mobile phase. Triethylamine was added in the mobile phase to prevent interaction of the basic alkaloids with acidic silanol groups of the silica gel packing material. Zones on these plates were visualized under UVGL-58 lamp (Analytica Jena, Upland, CA, USA) at 254/365 nm and then sprayed with Dragendoff’s reagent. The purity of the compounds was determined using RP-C18 column with analytical Shimadzu HPLC 2017 (Shimadzu Corporation, Kyoto, Japan)) with a photodiode array detector. A H_3_PO_4_/K_2_HPO_4_ buffer (pH 6.0) with mobile phase 4:6 (CH_3_CN/H_2_O) was used for the separation. Melting points were determined on a Mel Temp II apparatus (Merck, Darmstadt, Germany) and are uncorrected. Optical rotation was determined using an ADP430 digital polarimeter (Bellingham and Stanley, Tunbridge Wells, UK) at 20 °C using the sodium D line (589 nm). IR spectroscopy was performed on a Spectrum 2 FT-IR spectrometer (PerkinElmer, Waltham, MA, USA) with a UATR accessory. Mass spectrometry was performed using a Xevo G2-S mass spectrometer (Waters, Elstree, UK) equipped with an atmospheric solids analysis probe, in positive ion mode. ^1^H- and ^13^C-NMR spectra were recorded in CDCl_3_ at 500 and 125 MHz, respectively, with TMS as the internal reference, using an AVANCE III 500 MHz (^1^H) NMR spectrometer equipped with a BBFO probe (Bruker, Billerica, MA, USA). The chemical shifts (*δ*) of carbon and proton are in parts per million.

### 3.2. Plant Material

The stem barks of *V. africana* Stapf (Apocynaceae) were collected in Ndop, North West Region of Cameroon, in October 2014 by Dr. Wirmum Clare, Director of the Medicinal Foods and Plants, Bamenda. A voucher specimen of the plant, N° SCA887, was deposited at the Limbe Botanic Garden.

### 3.3. Extraction and Isolation

The fresh stem barks of *V. africana* were air-dried and ground into coarse powder. The powdered sample of the plant was macerated in methanol at room temperature for nine days (3 × 3 days). Filtration and concentration of the crude extract led to a dark greenish extract. The crude extract was subjected to silica gel normal phase open column chromatography and elution with a gradient of ethyl acetate in hexane. Repeated column chromatography and purification through Sephadex LH-20 and preparative TLC yielded the compounds **1**–**9**.

### 3.4. In Vitro Antimalarial Activity

#### 3.4.1. Isolation of *Onchocerca ochengi* Adult Worms

The isolation of *O. ochengi* adult worms was performed as described previously by Cho-Ngwa et al. [26]. Briefly, fresh pieces of umbilical cattle skin with palpable nodules bought from butchery in Douala, Cameroon were washed, drained and sterilized with 70% ethanol. The worms were carefully scraped out of the nodules as single masses and temporarily submerged in 1 mL complete culture medium, CCM [RPMI-1640 (Sigma-Aldrich, St. Louis, MO, USA) supplemented with 25 mM HEPES, 2 g/L sodium bicarbonate, 2 mM L-glutamine, 5% newborn calf serum (SIGMA, St. Louis, MO, USA), 150 units/mL penicillin, 150 µg/mL streptomycin and 0.5 µg/mL amphotericin B (SIGMA, St. Louis, MO, USA), pH 7.4)] using 24-well plates. The adult worms were incubated in the culture medium overnight in a CO_2_ incubator, during which period the male worms migrated out of the nodular masses. Only wells containing viable worms received treatment with test compounds. Damaged worms and worms from putrefied nodules were discarded. The viability of worms retained for the assay was ascertained by visual and microscopic examination of adult male worm motility using an inverted microscope.

#### 3.4.2. Mammalian Cells for Microfilarial Cultures and Cytotoxicity Studies

Monkey kidney epithelial cells (LLC-MK2) (ATCC, Manassas, VA, USA) were cultured at 37 °C in humidified air with 5% CO_2_ in a Hera Cell-150 incubator (Thermo Electron, Karlsruhe, Germany) until the cell layer was almost confluent. The cell suspension was dispensed into 96-well micro titer plates (200 µL/well) and kept in the incubator for 3–5 days for cells to grow and become fully confluent. These cells served as feeder layers for the Mf assays and were also used for cytotoxicity studies.

#### 3.4.3. Isolation and Culturing of *Onchocerca ochengi* Microfilariae

Briefly, Mf were prepared using the method described by Cho-Ngwa et al. [26], with slight modifications. A few skin snips were obtained from different locations and incubated in aliquots of culture medium for 15 min, after which the emergent Mf were qualified and quantified using an inverted microscope and standard atlases. The leftover pieces of skin were shaved, rinsed, sterilized with 70% ethanol and sliced into thin slivers. The slivers were incubated in CCM for 2 h, and the emergent highly motile *O. ochengi* Mf were concentrated by centrifugation. The Mf were transferred to 96-well micro titer plates (15 Mf/100 μL/well) containing a fully confluent LLC-MK2 cell layer in 100 μL of CCM. This was monitored for viability and sterility for 24 h before addition of test and control compounds.

#### 3.4.4. Preparation of *Loa*
*loa* Microfilariae

For the confirmation of the presence of *L. loa* Mf, a thick blood smear was prepared, stained with Giemsa and observed under a light microscope [26]. Ten milliliters of venous blood was collected in EDTA tubes and gently mixed. A portion of the blood was diluted in RPMI-1640 medium, and the Mf load was determined using an inverted microscope. The blood was then diluted according to the number of Mf present at initial count so as to obtain a total of 15 Mf/100 μL/well. After dilution, the Mf were transferred in a 96-well plate and monitored for 24 h before the addition of test and control compounds.

#### 3.4.5. Primary and Secondary Screens against *Onchocerca ochengi* Adult Worms

The compounds were tested on Mfs at a single concentration of 30 μM, in duplicate wells. The worm cultures containing the drug were incubated for 168 h (7 days), at 37 °C in 5% CO_2_ atmosphere. The female worms were removed and incubated in 500 μL of 0.5 μg/mL MTT for the last 30 min of incubation. Inhibition of formazan formation from MTT directly correlates with worm death. The worms were blotted on absorbent paper and observed visually for blue coloration against a white background. Scores based on activity were assigned, ranging from 100% inhibition of formazan formation giving a completely pale yellow worm, 90% inhibition giving only one or few spots of blue color seen on the worm, 75% inhibition where about 75% of the worm remained pale yellow, 50% where about 50% of the worm remained pale yellow, to 25% giving near total blue coloration, and 0% (inactive compound) for total blue color on worm. Auranofin at 10 μM, which had previously shown activity against *O. ochengi* adult worms and Mf [27], was used as a positive control, while negative control wells received the diluent, 2% DMSO only, previously shown to have no effect on parasite viability. The solvent DMSO (2%) was used as a negative control. In addition, the 2% DMSO concentration has been shown to be nontoxic to worms and LLCMK2 cells in vitro [26,36]. Furthermore, the toxicity of DMSO decreases with dilution [37].

Adult male worm motility was evaluated with the aid of an inverted microscope. Motility score was on a scale of 4 (vigorous or normal movement of whole worm, corresponding to 0% inhibition of worm motility), 3 (near normal movement of whole worm or 25% inhibition of worm motility), 2 (whole body of worm motile but sluggish, i.e., 50% inhibition of worm motility), 1 (only head or tail of worm moving, i.e., 75% inhibition of worm motility) and 0 (completely immotile worm, i.e., 100% inhibition of worm motility). A metabolite was considered active if there was a 100% inhibition of adult male worm motility, or moderately active for a motility inhibition of 50–99%, and inactive if the inhibition was less than 50%.

The pure compounds with 100% activity at primary screens were re-tested as described under primary screens, in order to determine the IC_50_ values. The IC_50_ assays were performed in triplicate and each experiment repeated for confirmation. The means of all activities at a concentration were calculated and used in the statistical analyses. GraphPadPrism version 6.0 (GraphPad Software, San Diego, CA, USA) was used to generate dose response curves from which the IC_50_ values were obtained.

#### 3.4.6. Primary and Secondary Screens against *Onchocerca ochengi* Microfilariae

The biological assays were conducted in duplicate in the 96-well microtiter plates. The Mf were incubated with the drug for 120 h in a total of 200 μL of medium. Mf viability was assessed by microscopy once every day and motility inhibition scores were recorded. The drug activity was determined using the data of the 5th day. Motility inhibition correlates to drug activity. The positive control drug was amorcazine at 30 μM, and negative control was the diluent (DMSO).

Compounds showing 100% activity in the primary screens were re-tested as described under primary screens to determine the IC_50_ values. All assays were repeated at least once. The selectivity index (SI) of each extract was calculated as the ratio of the IC_50_ of the extract on the mammalian cell (termed CC_50_) to the IC_50_ on parasites.

#### 3.4.7. Screens against *Loa*
*loa* Microfilariae

Ethical clearance for the study was obtained from the Cameroon National Ethics Committee, while patients freely gave their written consent to participate in the study before recruitment. Clients living in the Edea Health District were invited to the Edea District Hospital for free screening. All the compounds were screened against *L. loa* Mf and the IC_50_ values determined. This was performed according to the protocol used to screen extracts against *O. ochengi* Mf. All assays were repeated at least once for confirmation of results.

#### 3.4.8. Cytotoxicity Studies

Cytotoxicity of the pure compounds with anti-onchocercal activities was assessed on LLC-MK2 cells, microscopically, on day 5 of the Mf assay. Living cells were flattened out and attached to the culture plate, while dead cells were rounded up and detached from the plate. The IC_50_ values for these cells were estimated from the morphological deformation data.

#### 3.4.9. Statistical Analysis

The statistical significance of differences in means between the effects of pure compounds at various concentrations on parasites were determined by one-way analysis of variance (ANOVA), followed by Newman–Keuls multiple comparison tests. A value of *p* < 0.05 was considered significant. The data were analyzed using GraphPadPrism 6.

### 3.5. Molecular Modeling

In this study, it was unclear why the compounds exhibited anti-onchocercal activities. An attempt to explore the mode of action of these compounds and to explore the structure–activity relations led to docking all of them in the same active site as the reference compound in the study.

#### 3.5.1. Homology Modeling

The choice of the target thioredoxin reductase is based on the fact that the reference compound in the bioassays (Auranofin) is a known inhibitor of *Onchocerca volvulus* thioredoxin reductase [27]. Amino acid sequences of the target proteins were retrieved from several UniProt databases (https://www.uniprot.org/uniprot/I7IAK1) [38]. For the identification of the candidate template structures, a BLAST [39] search of the full-length target sequence was carried out using the RCSB Protein Data Bank [40]. The final template structures were chosen based on the sequence identity, the quality of the available structural data and the type of the bound ligand. Sequence identity and similarity of the templates and target proteins were calculated in MOE (Appendix A, Supplementary Data). The initial sequence alignment for modeling was performed in MOE software version 16.08 (Montreal, QC, Canada) [41]. The template structure and/or the initial alignment were manually adjusted before the modeling to address conserved structural features of the binding pocket. The homology modeling of the target protein catalytic domains was performed using the program MODELLER 9.11 (San Francisco, CA, USA) [42]. Choice of homology model was based on the DOPE score implemented in the modeler program.

##### Molecular Dynamic Simulation of the Generated Homology Model

The generated homology model was subjected to molecular dynamics (MD) simulations using the AMBER16 program (San Francisco, CA, USA) to evaluate its stability [35]. The *tleap* module in amber was used to prepare the system for MD simulations. Hydrogen atoms were added to all amino acid residues assuming a normal ionization state for all ionizable residues. The ff03.r1 force field was used to optimize the protein, while the tip3p for water model was used to solvate the entire system in an octahedral box [43]. The solute atoms and the borders of the box were separated by at least 10 Å.

The resulting system was then energy minimized in two steps: a 3000 step minimization (2000 cycles of steepest descent followed by 1000 cycles of conjugate gradient) with restraints on the protein atoms, while the solvent molecules and counter ions were free. This was followed by a further 4000 cycle minimization of the entire system (2000 cycles of steepest descent followed by 2000 cycles of conjugate gradient) without restraints to remove any steric clash in the initial geometry of the protein. Subsequently, restraint heating of the system using a force constant of 10 kcal/mol from 0 to 300 K was performed over a period of 100 ps. Finally, using the Particle Mesh Ewald method, we ran a 20 ns molecular dynamic simulation using a time step of 2 fs [44,45]. The system was kept at constant temperature (300 K) and pressure (1 bar) during the molecular dynamic simulation time. Distances and rmsd were calculated with respect to the input coordinates of the complex as a reference using *cpptraj*.

#### 3.5.2. Protein Preparation

Based on the generated homology models, the best model was selected (with regard to the lowest DOPE score) and subsequently used in this study. Protein preparation was performed using similar protocols previously reported by Simoben et al., 2018 [46]. Protein Preparation Wizard of Schrödinger software (New York, NY, USA) using the default settings was used to prepare the protein structure [47]. The Epik-tool (with the pH set at 7.0 ± 2.0) was used to apply bond orders, and hydrogen atoms and protonation of the heteroatom states were added using optimization of the H-bond network. The structure was finally subjected to a restrained energy minimization step using the Optimized Potentials for Liquid Simulations (OPLS) 2005 force field [48] and a root-mean-square-deviation (rmsd) of 0.3 Å for atom displacement for terminating.

#### 3.5.3. Ligand Dataset Preparation

Schrodinger software, v2017-u1 was used for the preparation of ligands for docking [49]. The Epik ionization method at biologically relevant pH (pH 7.0 ± 2.0) was used to generate all possible tautomers, as well as possible combinations of stereoisomers for molecules without well-defined stereochemistry. Additionally, the optimized integrated OPLS_2005 force field [48] was used to minimize all ligands. Finally, 30 conformers for each of the prepared ligand molecules were generated and each conformer output minimized using the settings of ConfGen [50,51].

#### 3.5.4. Ligand Docking and Scoring

Prepared protein and ligands were flexibly docked using the standardized Glide program procedure of Schrödinger’s software [49,51]. The receptor grid preparation for the docking procedure was carried out by assigning the center of the transferred ligand from the template into our homology model (FDA) as the centroid of the grid box. Glide docking program in Schrodinger software was subsequently used to dock the generated three-dimensional conformers of the prepared ligand. The GlideScore (GS) Standard Precision (SP) mode was used as the scoring function to rank the resulting binding poses [49,52]. A total of 5 poses per ligand conformer were included in the post-docking minimization step, and a maximum of 2 docking poses was generated for each ligand conformer.

#### 3.5.5. Rescoring Docked Poses by Binding Free Energy Calculations

This was performed via a complex minimization approach using the molecular mechanics energies combined with the generalized Born and accessible surface area (MM-GBSA) implicit continuum solvation [53,54] with the AMBER12EHT force field [55] implemented in MOE [41] to estimate the binding free energy (BFE) for the proposed poses after docking of each molecule. AMBER12EHT force field was used to fix partial charges for each of Protonate3D system [56] followed by a short minimization. Finally, an in-house script for MOE was used to estimated BFE for all the docked poses. Protein heavy atoms were tethered during complex minimization with a deviation of 0.5 Å (force constant (3/2) kT/(0.5)^2^).

## 4. Conclusions

A new bisindole iboga-vobasine type alkaloid named voacamine A (**1**) along with eight known compounds, voacangine (**2**), voacristine (**3**), coronaridine (**4**), tabernanthine (**5**), ibogaine (**6**), voacamine (**7**), voacorine (**8**) and conoduramine (**9**) were isolated from the stem barks of *Voacanga africana*. All compounds were tested for their anti-onchocercal activity against the *O*. *ochengi*. Compounds **1**–**4** and **6**–**8** were found to inhibit the motility of both the Mf and adult male worms of *Onchocerca ochengi*, in a dose-dependent manner, but were only moderately active on the adult female worms upon biochemical assessment at 30 μM drug concentrations. Their IC_50_ values are 2.49–5.49 µM for Mf and 3.45–17.87 µM for adult males. This study reports, for the first time, the efficacy of these alkaloids against Mf and adult *O. ochengi* worms. Molecular modeling led to the generation of a homology model for the *O. ochengi* thioredoxin reductase target. It was, however, unclear why the compounds exhibited anti-onchocercal activities. An attempt to explore the mode of action of these compounds and to explore the structure–activity relations led to docking all of them in the same active site as the reference compound in the study. Since the reference compound is known to bind to the aforementioned drug target, our rationale was to attempt an initial SAR study by exploring the putative binding of these compounds to the target in silico, then further suggest an in vitro study of the enzyme inhibition kinetics of these compounds against the target. Docking the receptor site could offer an explanation for the structure–activity relationships of the isolated compounds in terms of docking scores, binding free energies towards the drug target site, complex minimization energies and interactions with site chain residues.

## Figures and Tables

**Figure 1 molecules-26-00070-f001:**
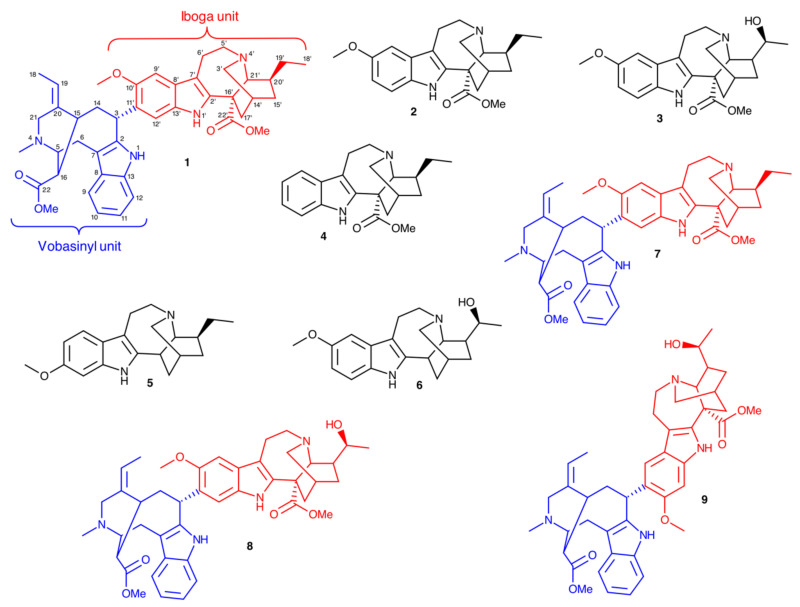
Chemical structures of compounds **1**–**9**. For the bisindoles (**1**, **7**–**9**), the vobasinyl and iboga subunits are highlighted in blue and red, respectively.

**Figure 2 molecules-26-00070-f002:**
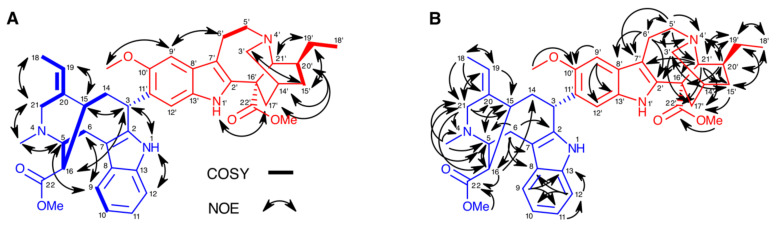
Key NOE (**A**) and HMBC (**B**) correlations in compound **1**. The vobasinyl and iboga subunits are highlighted in blue and red, respectively.

**Figure 3 molecules-26-00070-f003:**
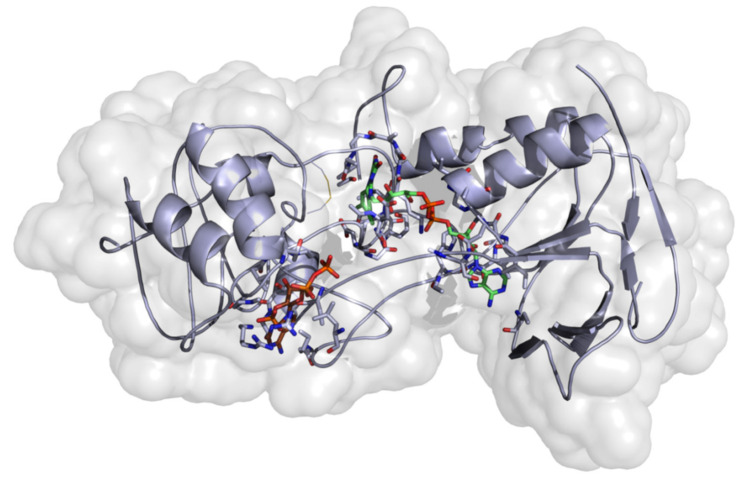
Homology model of thioredoxin reductase of *O. ochengi.*

**Figure 4 molecules-26-00070-f004:**
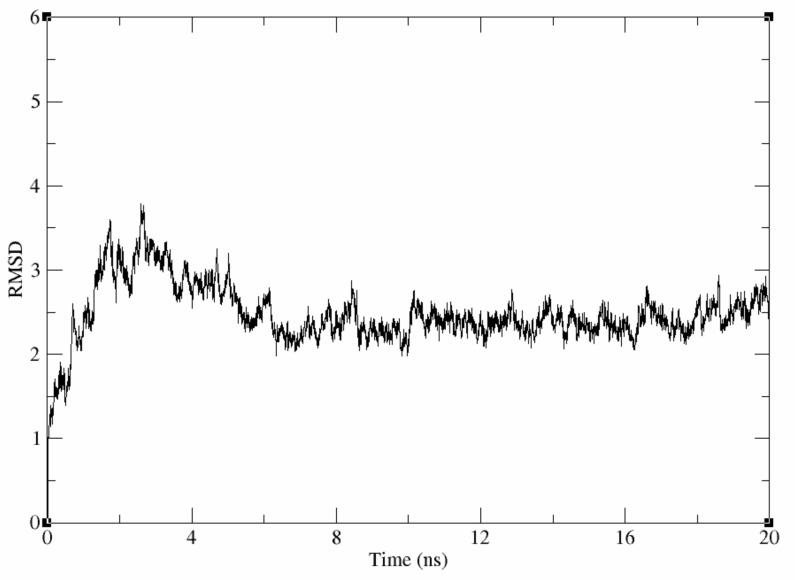
Root mean square deviation (rmsd) of generated homology sampled during 20ns molecular dynamics (MD) simulation with respect to the initial structure versus simulation time.

**Figure 5 molecules-26-00070-f005:**
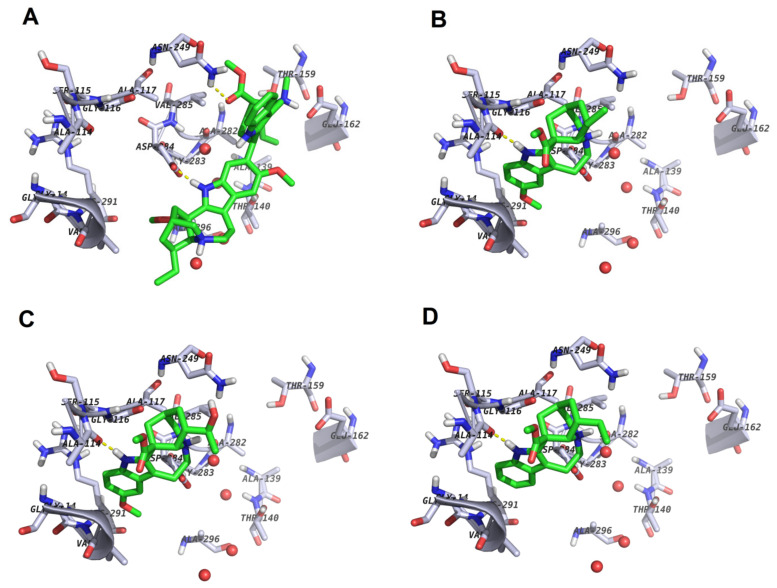
Binding interactions between the binding site with amino acid side-chains with the compounds; (**A**) compound **1** shown in green sticks, (**B**) compound **2** shown in green sticks; (**C**) compound **3** shown in green sticks, (**D**) compound **4** shown in green sticks.

**Table 1 molecules-26-00070-t001:** ^1^H (500 MHz) and ^13^C-NMR (125 MHz) assignments for compound **1** in CDCl_3._

	1						
**No.**	**δ_H_**	**δ_C_**	**δ_N_**	**No.**	**δ_H_**	**δ_C_**	**δ_N_**
1	7.74 (1H)		127.2	1′	7.49 (1H)		120.1
2		137.3		2′		137.2	
3	5.17 (1H)	37.3		3′	2.89 (1H) 2.74 (1H)	51.9	
4			40.8	4′			20.3
5	4.07 (1H)	60.0		5′	3.40 (1H) 3.17 (1H)	53.1	
6	3.51 (1H) 3.28 (1H)	19.5		6′	3.12 (1H) 2.99 (1H)	22.2	
7		110.0		7′		110.0	
8		129.9		8′		127.4	
9	7.57 (1H)	117.4		9′	6.95 (1H, s)	99.2	
10	7.08 (1H)	119.0		10′		150.9	
11	7.07 (1H)	121.6		11′		129.8	
12	7.07 (1H)	109.8		12′	6.76 (1H)	110.3	
13		135.8		13′		130.3	
14	2.58 (1H) 2.02 (1H)	36.3		14′	1.82 (1H)	27.3	
15	3.80 (1H)	33.6		15′	1.70 (1H) 1.11 (1H)	32.0	
16	2.75 (1H)	47.0		16′		54.9	
17				17′	2.50 (1H) 1.78 (1H)	36.4	
18	1.69 (3H)	12.3		18′	0.90 (3H)	11.6	
19	5.36 (1H)	118.8		19′	1.56 (1H) 1.45 (1H)	26.7	
20		137.8		20′	1.31 (1H)	39.0	
21	3.76 (1H) 2.95 (1H)	52.6		21′	3.53 (1H)	57.2	
22 CO		171.5		22′ CO		175.3	
22 OMe	2.48 (3H)	49.9		22′ OMe	3.67 (3H)	52.4	
4 NMe	2.63 (3H)	42.4		10′OMe	4.03 (3H)	56.1	

**Table 2 molecules-26-00070-t002:** Effect of isolated compounds on *O. ochengi* worms in primary screens.

* Compounds Tested (at 30 μM)	% Mf Motility Reduction after 24 h	% Adult Male Worm Motility Reduction after 24 h	% Adult Female Worm Death after 120 h
**1**	100	100	65
**2**	100	100	100
**3**	100	100	50
**4**	100	100	50
**5**	0	0	0
**6**	100	100	50
**7**	100	100	65
**8**	100	100	50
**9**	0	0	0
Ivermectin (10 μg/mL)	100	NA	NA
Auranofin (10 μM)	100	100	100
2% DMSO	0	0	0

* Auranofin was used as positive control for adult worm assay, while ivermectin, which is known not to kill adult worms, was used for Mf assay. Dimethyl sulfoxide (DMSO) 2% was used as negative control. Percentage adult female worm death corresponds to percentage inhibition of formazan formation. NA = Not applicable.

**Table 3 molecules-26-00070-t003:** IC_50_, IC_100_ and selectivity indices (SI) of isolates on *O. ochengi* microfilariae and adult worms, and monkey kidney epithelial cells (LLC-MK2) in secondary screens.

	Mf	Adult Male Worm	Adult Female Worm	Monkey Kidney Cells (LLC-MK2)	Mf	Adult Male Worm	Adult Female Worm	Monkey Kidney Cells (LLC-MK2)
	**1**				**2**			
**IC_50_** **(μM)**	3.69	4.45	-	≥30	5.49	9.07	-	≥30
**IC_100_** **(μM)**	7.38	8.90	>30	-	10	20	>30	-
**SI**	8.13	6.74			5.46	3.30	2.83	
	**3**				**4**			
**IC_50_** **(μM)**	4.34	8.07	-	≥30	4.21	8.68	-	≥30
**IC_100_** **(μM)**	8.68	16.14	>30	-	8.42	17.36	>30	-
**SI**	6.91	3.71			7.13	3.45		
	**6**				**7**			
**IC_50_** **(μM)**	4.72	9.07	-	≥30	2.49	3.45	-	≥30
**IC_100_** **(μM)**	9.44	18.14	>30	-	10	10	>30	-
**SI**	6.35	3.30						
	**8**							
**IC_50_** **(μM)**	2.49	3.45	-	≥30				
**IC_100_** **(μM)**	4.98	6.90	>30	-				
**SI**	12.04	8.69						

At 10 μM, amocarzine and FDA approved auranofin, a gold conjugated compound, previously shown to be a macrofilaricide and a current arthritis drug (positive control, [27]), also produced 100% inhibition of formazan formation in adult female worms at 120 h of incubation.

**Table 4 molecules-26-00070-t004:** Computed molecular descriptors for the assessment of the drug metabolism and pharmacokinetics (DMPK) profiles of the major isolated metabolites and the recommended range for 95% of known drugs.

Metabolite	^a^ #stars	^b^ CNS	^c^ MW (Da)	^d^ SASA	^e^ FOSA	^f^ FISA	^g^ Volume
**1**	9 *	1	706.9	1067.5 *	786.3 *	80.0	2140.6 *
**2**	0	2	368.5	626.1	480.8	28.0	1171.6
**3**	0	1	384.5	642.5	461.5	64.5	1191.3
**4**	0	2	338.4	588.5	387.1	28.0	1094.4
**5**	0	2	368.5	635.5	485.4	31.7	1175.4
**6**	0	1	384.5	642.4	461.5	64.5	1191.3
**7**	9 *	1	706.9	1057.7 *	781.6 *	71.7	2130.5 *
**8**	10 *	1	722.9	1091.1 *	769.0 *	119.3	2156.9 *
**9**	10 *	1	722.9	1089.4 *	768.4 *	115.5	2163.6 *
**Metabolite**	**^h^ HBD**	**^i^ HBA**	**^j^ logP**	**^k^ logS**	**^l^ logHERG**	**^m^ Caco-2**	**^n^ logBB**
**1**	1	9	7.2 *	−8.0 *	−8.66 *	26.8	0.2
**2**	0	4	4.5	−4.4	−5.11	1340.2	0.5
**3**	1	5	3.7	−4.1	−5.31	604.0	0.1
**4**	0	3	4.5	−4.3	−5.15	1341.8	0.6
**5**	0	4	4.5	−4.6	−5.29	1237.7	0.5
**6**	1	5	3.7	−4.1	−5.31	604.0	0.1
**7**	1	9	7.2 *	−7.8 *	−8.62 *	32.1	0.3
**8**	2	10	6.2	−7.5 *	−8.89 *	11.4	−0.4
**9**	2	10	6.2	−7.5 *	−8.86 *	12.4	−0.4
**Metabolite**	**^o^ MDCK**	**^p^ logK_p_**	**^q^ #metab**	**^r^ logK_HSA_**	**^s^ PHOA**	**^t^ Ro5**	**^u^ Ro3**
**1**	70.4	−6.1	10 *	2.49 *	92.7	2 *	2 *
**2**	282.3	−3.4	2	0.89	50.3	0	0
**3**	133.9	−4.0	3	0.63	69.0	0	0
**4**	268.8	−3.3	1	0.91	42.0	0	0
**5**	836.0	−3.5	2	0.90	50.1	0	0
**6**	340.4	−4.0	3	0.63	69.1	0	0
**7**	24.1 *	−5.9	10 *	2.47 *	95.1	2 *	2 *
**8**	9.2 *	−6.7	10 *	2.09 *	118.8	2 *	3 *
**9**	26.4	−6.6	10 *	2.11 *	118.1	2 *	3 *

* Property which falls outside the recommended range for 95% of known drugs; ^a^ number of computed properties which fall outside the required range for 95% of known drugs (recommended range 0 to 5); ^b^ activity in the central nervous system in the scale −2 (inactive) to +2 (active); ^c^ molar weight (range for 95% of drugs: 130–725 Da); ^d^ the solvent accessible surface area (recommended range 300.0 to 1000.0 Å^2^); ^e^ The hydrophobic component of the solvent accessible surface area (recommended range 0.0 to 750.0 Å^2^); ^f^ the hydrophilic component of the solvent accessible surface area (recommended range 7.0 to 330.0 Å^2^); ^g^ total volume of molecule enclosed by solvent-accessible molecular surface, in Å^3^ (probe radius 1.4 Å) (range for 95% of drugs: 500 to 2000 Å^3^); ^h^ number of hydrogen bonds donated by the molecule (range for 95% of drugs: 0 to 6); ^i^ number of hydrogen bonds accepted by the molecule (range for 95% of drugs: 2–20); ^j^ logarithm of partitioning coefficient between *n*-octanol and water phases (range for 95% of drugs: −2 to 6.5); ^k^ the predicted aqueous solubility, with S in mol/dm^3^ (range for 95% of drugs: −6.5 to 0.5); ^l^ predicted IC_50_ value for blockage of HERG K+ channels (concern <−5); ^m^ predicted apparent Caco-2 cell membrane permeability in Boehringer–Ingelheim scale, in nm/s (range for 95% of drugs: <5 low, >500 high); ^n^ logarithm of predicted blood/brain barrier partition coefficient (range for 95% of drugs: −3.0 to 1.0); ^o^ the predicted apparent MDCK permeability in nm/s (<25 poor, >500 great); ^p^ the predicted skin permeability (range for 95% of drugs: −8.0 to −1.0); ^q^ number of likely metabolic reactions (range for 95% of drugs: 1–8); ^r^ Logarithm of predicted binding constant to human serum albumin (range for 95% of drugs: −1.5 to 1.5); ^s^ the predicted percentage human oral absorption (>80% high, <25% poor); ^t^ number of violations of Lipinski’s ”Rule of Five” (recommended maximum of 4); ^u^ number of violations of Jorgensen’s ”Rule of Three” (recommended maximum of 3).

**Table 5 molecules-26-00070-t005:** Docking scores of the isolated compounds and their moieties towards the *O. ochengi* thioredoxin reductase target and minimization energies of the protein–ligand complexes (including electronic, solvation and van der Waals components).

Compound	Minimization Energy (Amber12, kcal/mol)	Docking Score (SP *, kcal/mol)
**1**	−30.48	−4.25
**1a** ^a^	−32.02	−5.07
**1b** ^b^	−32.33	−4.89
**2**	−32.01	−5.07
**3**	−28.94	−5.26
**4**	−26.79	−4.54
**5**	−28.72	−4.80
**6**	−28.88	−5.26
**7**	−30.43	−4.25
**7a** ^a^	−31.99	−5.08
**7b** ^b^	−21.83	−4.18
**8**	−28.41	−4.73
**8a** ^a^	−30.34	−5.28
**8b** ^b^	−21.91	−4.18
**9**	−40.53	−5.20
**9a** ^a^	−29.65	−5.03
**9b** ^b^	−21.59	−4.18
Auranofin	−47.38	−6.37

* Standard Precision; ^a^ the iboga unit; ^b^ the vobasinyl unit.

**Table 6 molecules-26-00070-t006:** Binding free energy (kcal/mol) and its components of docked compounds to the protein target.

Compound	ΔG_bind_	E_ele_	E_vdw_	E_sol_
**1**	−38.59	−94.21	−33.61	89.23
**1a**	−23.65	−92.54	−18.71	87.6
**2**	−33.95	−75.89	−29.02	70.96
**3**	−34.01	−68.58	−29.51	64.08
**4**	−29.32	−70.64	−25.32	66.64
**5**	−26.57	−111.25	−23.41	108.08
**6**	−27.74	−63.78	−22.78	58.82
**7**	−40.47	−81.75	−36.67	77.95
**7a**	−24.67	−39.34	−21.48	36.15
**9**	−29.83	−155.19	−24.87	150.22
**9b**	−28.49	−56.25	−23.83	51.59

## Data Availability

Data is contained within the article or supplementary material.

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
