# Peer review of "Alkaloids with Anti-Onchocercal Activity from Voacanga africana Stapf (Apocynaceae): Identification and Molecular Modeling"

_molecules, 2020, doi:10.3390/molecules26010070_

Round 1

Reviewer 1 Report

In the previous revision, I included two minor comments for this paper: one of these was addressed (about Table 4), the other was apparently not (about the discussion of the binding modes). I have the strange feeling that I received a wrong version of the paper, since the text about the binding modes was exactly the same as in the earlier version (except for changing the numbering of Figure 4 to Figure 5), even though the authors have noted that they've modified it. Even more strangely, the rest of the binding modes (Figure S29) and the structure of auranofin (Figure S28) have now disappeared from the supplementary information, even though the main text still refers to these.

I would ask the authors and the editor to double-check what might've gone wrong.

The paper should be publishable after we sort out this (presumably technical) issue.

Author Response

Response:

This has been checked and corrected in the updated manuscript.

The Figure S28 has been included, while comments about Figure S29 have been removed from the original manuscript text.

The ethical clearance document has also bee attached in the re-submission.

Reviewer 2 Report

The authors fulfilled all the suggestions and modifications which would improve their paper. They also better explained their target's choice and added this important information in two different sections of their manuscript, highlighting that the molecular modelling studies were performed starting from a hypothesis and not from a known target. This makes their approach suitable and, however still performed over a hypothetic target, their results could be important to other research groups. The molecular modelling methodologies are much more robust after some adequations. They also provided a better explanation about the employed descriptors which, enhanced the understanding of their results. I still think that 1% DMSO is the maximum concentration of such solvent to the biological activity, but as I am not expert in this biological assay issue, I can accept the references in which the authors based their discussions. For these reasons, I consider the paper is now suitable for publication.

Author Response

Response: Thanks for the comments.

Round 2

Reviewer 1 Report

I'm afraid the authors have misunderstood me about the binding poses of the other compounds (formerly Figure S29). These are necessary for the manuscript, since the authors still discuss these in the section "2.3.5 Structure-Activity Relationships" (e.g. comparing the binding poses of compounds 1 vs 1a, or 7 vs 7a), please include these binding poses in the supplementary information (I was just puzzled why were they removed at all).

Also, page 13, paragraph 3, last sentence is still not corrected: "This could be because this part of the molecule does not interact with the aforementioned residues, the two compounds (1 and 7) almost binding in the same way (yellow structures of Figures 5A and D)."

This sentence should correctly refer to either:

  • compounds 1 and 4 (Figure 5A and D)
  • or compounds 1 and 7 (Figure 5A and Figure S29)
  • or perhaps compounds 2, 3 and 4 (Figure 5B, C and D), whose binding poses are much more similar than those of 1 and 4.

Please decide which one you meant and correct this sentence.

Author Response

I'm afraid the authors have misunderstood me about the binding poses of the other compounds (formerly Figure S29). These are necessary for the manuscript, since the authors still discuss these in the section "2.3.5 Structure-Activity Relationships" (e.g. comparing the binding poses of compounds 1 vs 1a, or 7 vs 7a), please include these binding poses in the supplementary information (I was just puzzled why were they removed at all).

Response: These have been included in the Supplementary information as Figures S29 and S30.

Also, page 13, paragraph 3, last sentence is still not corrected: "This could be because this part of the molecule does not interact with the aforementioned residues, the two compounds (1 and 7) almost binding in the same way (yellow structures of Figures 5A and D)."

Response: These have been corrected.

This sentence should correctly refer to either:

  • compounds 1 and 4 (Figure 5A and D)
  • or compounds 1 and 7 (Figure 5A and Figure S29)
  • or perhaps compounds 2, 3 and 4 (Figure 5B, C and D), whose binding poses are much more similar than those of 1 and 4.

Please decide which one you meant and correct this sentence.

Response: These have been corrected.